# 3D Scout Scans Using Projection Domain Denoising

**Mikhail Bortnikov**[1]                                              MIKHAIL.BORTNIKOV@PHILIPS.COM
**Frank Bergner**[2]                                                    FRANK.BERGNER@PHILIPS.COM
**Alexey Chernyavskiy**[1]                                        ALEXEY.CHERNYAVSKIY@PHILIPS.COM
**Kevin M. Brown**[3]                                              KEVIN.M.BROWN@PHILIPS.COM
**Noel Black**[3]                                                        NOEL.BLACK@PHILIPS.COM
**Michael Grass**[2]                                                  MICHAEL.GRASS@PHILIPS.COM

[1] *Philips Innovation Labs RUS, Moscow, Russia*

[2] *Philips Research Hamburg, Germany*

[3] *Philips CT Research & Advanced Development, Cleveland, OH, USA*

**Editors:** Under Review for MIDL 2021

## Abstract

Low dose 2D scouts, also known as topograms, are commonly used for CT scan planning. Although 3D CT volumes can provide more valuable information for the selection of the scan range and parameters, the very low X-ray dose used during scout scan acquisitions results in artefacts requiring effective denoising techniques to make them useful. This has proved challenging for traditional denoising algorithms. We propose a projection domain denoiser based on a convolutional neural network (CNN), which provides improved image quality even at ultra-low dose levels. We compare two CNNs operating on two data representations, a conventional line integral data and raw photon counts, which have different quantitative properties and dynamic ranges. The results show that the two denoising strategies give rise to different properties of reconstructed images and that both projection CNNs are effective for ultra-low dose CT denoising.

**Keywords:** Computer Vision, Medical Imaging, Image Denoising, Deep Learning, Computed Tomography

## 1. Introduction

A 3D scout is a CT scan captured with the X-ray dose of 2D scout, which is approximately 1% of the full CT dose. The usage of 3D data for planning allows accurate scan planning and optimization of patient dose during the diagnostic scan. E.g., dedicated planning for the position of certain soft-tissue organs is difficult if these do not have enough contrast to be visible in the 2D X-ray images.

CT aims at reconstructing an unknown X-ray attenuation map $\mu(.)$ of the volume. The attenuation of X-ray photons on a ray through the volume follows the Lambert–Beer's Law, and in combination with photon statistics the number of photons captured by the detector can be modelled with the following distribution:

$$I_i \sim Poisson\{I_{i,0}e^{-p_i}\} = Poisson\{I_{i,0}e^{-\int_0^\infty \mu(\eta)d\eta}\} \tag{1}$$

where $p_i$ is the line-integral for the i-th detector pixel, $I_{i,0}$ are the photons from the tube towards the i-th detector pixel without attenuation.

Recently, several attempts have been made at creating a low dose 3D scout scanning protocol. They include a strategy to emit a lower number of photons while trying to keep high image quality (Yin et al., 2015), reconstructing images with stronger regularization (Gomes et al.) or applying a denoiser to the resulting images.

In this work we focus on CT denoising. Once the dose is getting very low, certain parts of the detector will suffer from *photon starvation*, i.e. there will be no photons measured. This effect results in severe artefacts in the reconstructed image. By operating on projections one can inpaint the regions where photon information is missing. We compare two CNN denoisers trained using line-integrals data and raw photons data ($p_i$ and $I_i$ in eq. 1). Our hypothesis is that the higher dynamic range of the raw photons data (from 0 to $10^6$) is beneficial for denoising task, because the deviation between signal and noise is more prominent. We use the TV-denoiser (Brown et al., 2011) as a baseline for comparison.

## 2. Material and Method

X-ray projections from multiple rays acquired during a CT scan are stacks of 2D data, obtained by rotating an X-ray source around the patient, and registered by a two-dimensional sensor array. Given high-dose projections of patients, we simulate low dose projections equivalent to 100x reduction of X-ray intensity using a known Poisson noise model (Žabić et al., 2013). For training and validation of our CNN we used a dataset of high dose CT projections of patients. Low dose projections equivalent to 100x reduction of X-ray intensity were generated from them using low dose simulation algorithm. We extracted overlapping patches of $64 \times 64$ from 160000 projections for training, and 20000 projections were set for validation. Two separate CT volumes were set as a holdout.

We train 16-layer CNN to predict residuals of noise, which are then subtracted from the noisy low dose projections to obtain an estimate of the high-dose projections. In a typical CT scan there is significant correlation between adjacent projections, which we utilized in form of 2.5D input. We removed biases from the convolutional layers and removed batch normalizations, therefore, our model became more robust to different noise levels. The CNN was trained using Adam optimizer. Mean squared error (MSE) was used as loss function.

The same CNN architecture was trained on line integral and raw photon data, while the latter were brought prior to image reconstruction back to the line integral domain. After network inference on low dose 3D scout helical scan data, image reconstruction was applied to generate volumetric data sets with sufficient quality to enable three-dimensional scan planning. For validation, we measured MSE, SSIM (structural similarity index), and conducted visual inspection of both the projections and the reconstructed images.

## 3. Results and Discussion

The CNNs significantly outperforms the TV-denoiser both in projection and image domain. Despite the fact that pixel-based image quality metrics are better for projections and images denoised by the CNN that was trained on line integrals data (Table 1), visual inspection of restored images shows that the photon based network restores the same amount of information (Figure 1). Smoother and less noisier projections of line-integrals CNN correspond to smooth images with loss of resolution in the edges, while the noisier output of photon

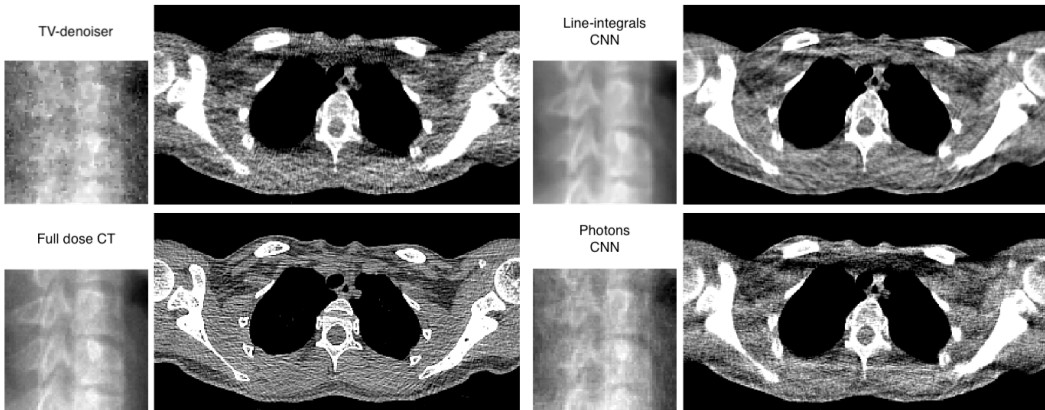

Figure 1: Pairs of denoised projections and corresponding reconstructed images (L/W = 50/500). Left column: TV-denoiser, full dose CT. Right column: line-integrals CNN, photons CNN

| Denoiser | Projections (full scan) | | Images (full volume) | |
|---|---|---|---|---|
| | SSIM | MSE | SSIM | MSE, $HU^2$ |
| Low dose | 0.435 | 0.120 | 0.275 | 38007 |
| TV-denoiser | 0.635 | 0.022 | 0.332 | 3646 |
| Line-integrals CNN | **0.943** | **0.004** | **0.383** | **1733** |
| Photons CNN | 0.803 | 0.014 | 0.338 | 2586 |

Table 1: Denoising performance comparison on projections and on reconstructed images

based CNN results in sharper image with additional artefacts. While the potential to use CNN for projection based denoising in extremely low dose situation could be demonstrated, future work to achieve improved results may include the use of slightly less aggressive dose reduction as well as alternative loss functions and architectures.

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
