# OpenReview forum: "3D Scout Scans Using Projection Domain Denoising"
_MIDL.io/2021/Conference/Short — MIDL 2021 Poster_

### Official Review · Reviewer_RUNV · 2021-04-30

**Confidence:** 3
**Final Rating:** 2

**Summary:**

The authors address the problem of restoring (denoising) scout scan acquisitions employing low X ray dose. Using such low doses leads to artifacts caused by photon starvation. Traditional (non-learning based) denoising techniques struggle to correct for these introduced artifacts. In this work, the authors explore training a CNN in a supervised fashion in order to denoise a scout scan acquisition. As training data, they simulate low dose acquisitions to be used as input data. They demonstrate improved performance in terms of SSIM and MSE metrics with their approach on the held-out data.

**Strengths:**

- The authors present a deep learning solution to restoring low dose scout scan acquisitions. Their two approaches score higher on evaluation metrics in the held out data, as compared to one non-learning based denoising solution


**Weaknesses:**

- It was not clear to me what exactly is the network architecture -  Is it a Resnet16 or rather a custom architecture? What is the filter size etc?
- Results seem to be arrived at after one run (it would be recommended to run the experiments a few times and show evaluation scores with variance.)


**Deanonymize Review:**

yes

**Detailed Comments:**

- The paper lacks explaining a general motivation behind 3d scout scans.  Adding a couple of lines prior to introducing the problem would significantly improve readability
- The paper refers to the "low dose simulation algorithm". A reference to a relevant work employing this, could be included.
- Adding motivation for specifically choosing the TV-denoiser as a baseline method could be included
- HU = Hounsfield Unit could be mentioned in the paper
- "We removed biases from the convolution layer ..." - Please add a reference supporting the line of action for making CNNs robust to noisy inputs (perhaps [Denoising with Bias-Free CNNs](https://openreview.net/pdf?id=HJlSmC4FPS)?)
- Comparison to supervised baselines such as [CARE](https://www.nature.com/articles/s41592-018-0216-7) could be explored?
- Comparison to a weakly supervised approach such as  [N2N](https://arxiv.org/pdf/1803.04189.pdf) could be explored - i.e. is it necessary to have target high dose acquisitions during training?
- Since holes caused by photon starvation seems to be a common artifact, would training an inpainting (instead of a denoising) network help?
- A discussion explaining why Line Integral CNN scored better than Photons CNN would be welcome.

**Justification Of The Rating:**

This paper provides a DL-based solution for restoring low dose scout scans. The task is quite interesting and the authors appear to have arrived at a fair solution.  I rate this as a weak reject primarily because the scores on evaluation metrics seemed to have been arrived after only one run. Other than that, the writing is a bit muddled and can be improved in my opinion. A few more validation studies would also make this work more robust.

**Paper Type:**

both

**Special Issue:**

no

---

### Official Review · Reviewer_dsZf · 2021-05-07

**Confidence:** 4
**Final Rating:** 4

**Summary:**

The authors present work designed to denoise 3D scout scans for improved visualization.  Two different CNNs are used, operating on two different data representations, namely the line integral data and the raw photon counts.  The CNN models operate on projections from which the image volume can be later reconstructed.

**Strengths:**

The paper is generally well written with a good explanation of the motivation for the task and mentions of recent literature.  The methods are explained with as much detail as is reasonable and the results are clear with both a figure and a table.  I am not extremely familiar with denoising literature but if this work is as novel as the authors claim then I believe it could represent a very nice advancement in the field.

**Weaknesses:**

The description of the dataset could be improved.  I would like to know how many patients, volumes, projections for each of training/validation and test sets, and I would like to know how the split into these sets was made.   The choice of baseline method should be justified, it is not clear to me whether this represents the current state of the art or is a random choice.

**Deanonymize Review:**

no

**Detailed Comments:**

The top row of Table 1 is slightly confusing to me.  I am not clear on what is meant by "projections (full scan)" versus "Images (full volume)".  I assumed that projections were the 2D projections so it is not clear what "full scan" means in that context.

**Justification Of The Rating:**

The paper represents novel work as far as I am aware and presents interesting and convincing results which outperform the baseline method.   The work is well explained and I believe it is an interesting study for the MIDL community.

**Paper Type:**

validation/application paper

**Special Issue:**

no

---

### Meta-Review · Area_Chair_r4VA · 2021-05-07

**Recommendation:** Accept (Poster)
**Confidence:** 5

**Metareview:**

While the AC agrees that the experiments appear to be in an early state, the results seem promising and worth discussing at MIDL.

---

### Decision · Program_Chairs · 2021-05-11

Accept (Poster)